# Histamine H_3_ Receptor Ligands—KSK-59 and KSK-73—Reduce Body Weight Gain in a Rat Model of Excessive Eating

**DOI:** 10.3390/ph14111080

**Published:** 2021-10-25

**Authors:** Kamil Mika, Małgorzata Szafarz, Marek Bednarski, Gniewomir Latacz, Sylwia Sudoł, Jadwiga Handzlik, Krzysztof Pociecha, Joanna Knutelska, Noemi Nicosia, Katarzyna Szczepańska, Kamil J. Kuder, Katarzyna Kieć-Kononowicz, Magdalena Kotańska

**Affiliations:** 1Department of Pharmacological Screening, Jagiellonian University Medical College, Medyczna 9, PL 30-688 Cracow, Poland; kamil.mika@doctoral.uj.edu.pl (K.M.); marek.bednarski@uj.edu.pl (M.B.); joanna.1.knutelska@uj.edu.pl (J.K.); noemi.nicosia92@gmail.com (N.N.); 2Department of Pharmacokinetics and Physical Pharmacy, Jagiellonian University Medical College, Medyczna 9, PL 30-688 Cracow, Poland; malgorzata.szafarz@uj.edu.pl (M.S.); krzysztof.pociecha@uj.edu.pl (K.P.); 3Department of Technology and Biotechnology of Drugs, Faculty of Pharmacy, Jagiellonian University Medical College, Medyczna 9, PL 30-688 Cracow, Poland; gniewomir.latacz@uj.edu.pl (G.L.); s.sudol@doctoral.uj.edu.pl (S.S.); jadwiga.handzlik@uj.edu.pl (J.H.); szczepanskatarzyna@gmail.com (K.S.); kamil.kuder@uj.edu.pl (K.J.K.); katarzyna.kiec-kononowicz@uj.edu.pl (K.K.-K.); 4Foundation “Prof. Antonio Imbesi”, University of Messina, Piazza Pugliatti 1, 98122 Messina, Italy; 5Department of Chemical, Biological, Pharmaceutical and Environmental Sciences, University of Messina, Viale Palatucci, 98168 Messina, Italy; 6Department of Medicinal Chemistry, Maj Institute of Pharmacology Polish Academy of Sciences, Smętna 12, PL 31-343 Cracow, Poland

**Keywords:** excessive eating model, obesity, histamine H_3_ receptor ligands, (4-pyridyl)piperazine derivatives

## Abstract

Noting the worldwide rapid increase in the prevalence of overweight and obesity new effective drugs are now being sought to combat these diseases. Histamine H_3_ receptor antagonists may represent an effective therapy as they have been shown to modulate histamine synthesis and release and affect a number of other neurotransmitters (norepinephrine, acetylcholine, γ-aminobutyric acid, serotonin, substance P) thus influencing the food intake. Based on the preliminary studies determining affinity, intrinsic activity, and selected pharmacokinetic parameters, two histamine H_3_ receptor ligands were selected. Female rats were fed palatable food for 28 days and simultaneously administered the tested compounds intraperitoneally (i.p.) at a dose of 10 or 1 mg/kg b.w./day. Weight was evaluated daily and calorie intake was evaluated once per week. The plasma levels of cholesterol, triglycerides, leptin, adiponectin, ghrelin, corticosterone, CRP and IL-6 were determined at the end of experiment. The glucose tolerance test was also performed. To exclude false positives, the effect of tested compounds on spontaneous activity was monitored during the treatment, as well as the amount of consumed kaolin clay was studied as a reflection of possible gastrointestinal disturbances comparable to nausea. The histamine H_3_ receptor antagonists KSK-59 and KSK-73 administered i.p. at a dose of 10 mg/kg b.w. prevented weight gain in a rat model of excessive eating. They reduced adipose tissue deposits and improved glucose tolerance. Both compounds showed satisfying ability to penetrate through biological membranes determined in in vitro studies. Compound KSK-73 also reduced the caloric intake of the experimental animals what indicates its anorectic effect. These results show the pharmacological properties of histamine H_3_ receptor antagonists, (4-pyridyl)piperazine derivatives, as the compounds causing not only slower weight gain but also ameliorating some metabolic disorders in rats having the opportunity to overeat.

## 1. Introduction

Obesity, nowadays, is a serious social, health and economic problem. Furthermore, it is well-established that obesity can progressively cause and/or exacerbate a wide spectrum of diseases: ischemic heart disease, hypertension, diabetes, diseases of the digestive system such as gallstone disease and nonalcoholic fatty liver disease. It can also increase the risk of developing colorectal, gallbladder, pancreas or kidneys cancers [1]. The number of people struggling with obesity has increased threefold since 1980. Statistical studies show that up to a third of the world’s population is currently overweight or obese. This trend will likely continue to fuel the global obesity epidemic for decades to come, worsening the population health and posing a challenge for many research groups and the pharmaceutical industry to seek for a new effective pharmacotherapy [2].

However, the pharmaceutical industry is experiencing a crisis in terms of introducing new drugs for the treatment of obesity. The main reasons for this are frequent side effects resulting from the use of anti-obesity medications, but also the low effectiveness of individual drugs. Five medications are currently approved by the Food and Drug Administration (FDA) for chronic weight management: phentermine/topiramate, bupropion/naltrexone, orlistat, liraglutide and semaglutide. Often, however, these drugs induce only a small degree of weight loss and are frequently discontinued by the patients due to the adverse reactions accompanying their use [3].

The small number of effective and safe drugs, and the prevalence of obesity in the world are a challenge for scientists and the pharmaceutical industry to look for new compounds that may be useful in the pharmacotherapy of obesity. One of the new directions in the treatment of obesity is use of histamine H_3_ receptor ligands. Histamine H_3_ receptors are located in the central nervous system and modulate histamine synthesis and release. Histamine, in turn, plays a significant role in eating behavior because it causes a loss of appetite and is considered a satiety signal released during the food intake [4]. Additionally, based on previous studies, it has been proven, that histamine H_3_ receptors affect release of a number of neurotransmitters and peptides, such as noradrenaline, acetylcholine, γ-aminobutyric acid, serotonin, and substance P, indirectly affecting food intake or physical activity [5,6]. Furthermore, histamine influences peripheral metabolism, responsible for lipolysis of white adipose tissue [5]. It has been reported previously that antagonists and inverse agonists of histamine H_3_ receptors are also capable of reducing the level of triglycerides in plasma [7], which is important since lipid abnormalities associated with obesity might highly increase the risk of developing the type II diabetes and cardiovascular system diseases.

Looking at the location of histamine receptors and their physiological function, it is believed that their antagonists and inverse agonists could be useful in the treatment of obesity. In the preclinical studies it was found that histamine H_3_ receptor antagonists reduce body weight. Several studies also showed that the blockade of histamine H_3_ receptor significantly reduces food intake by animals [8,9,10,11,12].

To date, two compounds with antagonistic activity against the histamine H_3_ receptor have entered clinical trials for the treatment of obesity. A selective histamine H_3_ receptor antagonist, compound HPP404, has entered phase II clinical trials for the treatment of overweight and obesity. Phase II clinical trials with SCH-497079 were recently completed, however due to the side effects, this compound was removed from further studies [13]. Betahistine, a potent histamine H_3_ receptor antagonist and H_1_ histamine receptor agonist, has also entered phase II clinical trials with an indication in the treatment of obesity [14]. Our group recently reported the positive effect of this drug on the selected metabolic parameters as well as the amount of tasty food consumed by the animals in the overeating model [15], emphasizing that the search for new indications for already approved drugs is beneficial. The selective histamine H_3_ receptor antagonist A-331440 caused weight loss in mice fed high fat diet, with the highest dose of A-331440 reducing body weight to a level similar to that observed in the low-fat diet fed mice. This compound also affected the biochemical parameters: lowered leptin levels, and normalized insulin tolerance. Unfortunately, it proved to be genotoxic and was not included in clinical trials [4,16].

Although it is well known that obesity can result from eating high-calorie foods or very large portions, some people do not refrain from excessive eating and as a result develop obesity. The concept of inhibiting food intake by histamine H_3_ receptor ligands as a tool to combat the plague of overweight and obesity in an overeating society seems to be appropriate and research aimed at assessing possible activity and initial safety of subsequent compounds is extremely important. In this work, we show the activity of two histamine receptor ligands (KSK-59 and KSK-73) in the excessive eating model, their effect on food intake, body weight and selected metabolic parameters. We also compare pharmacological effects of the investigated compounds to the combination of drugs currently used in the treatment of obesity: bupropion/naltrexone. To exclude false-positive results, we also determined: (a) the influence of selected compounds on spontaneous activity, (b) whether the compounds cause stress, (c) whether they cause gastrointestinal disorders comparable to nausea, and d) whether inflammation might occur during chronic administration. Moreover, we discuss how these compounds differ from the others discovered in our laboratory so far [11,12]. In addition, in in vitro studies, we show that these compounds are antagonists of H_3_ histamine receptor, and for the KSK-59 compound we also report the results of its membrane permeability.

## 2. Results

### 2.1. The Intrinsic Activity at Histamine H_3_ Receptor

The intrinsic activity of all tested compounds towards histamine H_3_ receptor was examined using two commercial methods. Since both methods differ in the type of cells used, the reaction environment, the method of signal transduction and its detection, some differences in the obtained IC_50_ values are expected and were observed even for the standard agonist, i.e., (R)-alpha-methylhistamine.

Compound KSK-59 proved to be a stronger histamine H_3_ receptor antagonist with the IC_50_ value of just over 3 nM determined in both methods. Compound KSK-73 showed slightly less activity in blocking histamine H_3_ receptor, and its IC_50_ value was 10–20 nM depending on the method used.

The activity of the tested compounds was similar or slightly weaker compared to the reference compounds (clobenpropit and thioperamide), standard histamine H_3_ receptor antagonists (Table 1, Figure 1a,b).

### 2.2. Permeability Profile

The parallel artificial membrane permeability assay (PAMPA) was used to estimate the ability of KSK-59 to passively penetrate lipid membranes. The same method was used previously for KSK-73 testing [17]. Results of PAMPA allow also to estimate the ability of tested compounds to penetrate blood-brain barrier (BBB) by passive transport. The calculated permeability coefficient (*Pe*, cm/s) for KSK-59 was slightly lower than the one determined for KSK-73, and both were lower than that of highly permeable caffeine. However, PAMPA plate’s manufacturer breakpoint for permeable compounds is *Pe* ≥ 1.5 × 10^−6^ cm/s according to the PAMPA test protocol. Therefore, in comparison to the low-permeable reference (norfloxacin) and the manufacturer breakpoint, both tested compounds showed satisfying ability to penetrate through biological membranes, including BBB, by passive transport (Table 2).

### 2.3. Influence of the Tested Compounds or Bupropion/Naltrexone on Body Weight

Control animals receiving palatable feed gained significantly more weight than control animals receiving standard feed. Looking at the average weight of rats on the last day of experiment, the rats fed standard feed gained 55% compared to their initial body weight, while the rats fed palatable feed gained 67%.

Animals fed palatable feed and receiving intraperitoneal (i.p.) injections of KSK-59 or KSK-73 at a dose of 10 mg/kg b.w. or bupropion/naltrexone combination gained less weight than the control group fed palatable diet and receiving i.p. vehicle (Figure 2 and Figure 3a,b). It is noteworthy that the body weight gain of animals treated with KSK-59 or KSK-73 was similar to that observed in animals receiving the reference compound. Compound KSK-73 administered at a dose of 10 mg/kg b.w. had even more potent effect than the bupropion/naltrexone combination, and slowed down the rats’ body weight gain already after the first administration. Test compounds administered at a dose of 1 mg/kg b.w. to rats fed palatable feed did not affect their body weight compared to the control rats fed the same feed. 

### 2.4. Influence of the Tested Compounds or Bupropion/Naltrexone on Fat Pads

Animals fed palatable feed and receiving i.p. vehicle had a statistically significantly higher number of fat pads than animals fed standard feed. Significantly fewer fat pads in the peritoneal cavity were noted in animals receiving i.p. injections of KSK-73 or bupropion/naltrexone compared to the group receiving the vehicle and fed the same palatable feed (Figure 3c). It is noteworthy that animals receiving i.p. injections of KSK-73 had even fewer fat pads than animals treated with the reference compound. 

### 2.5. Influence of the Tested Compounds or Bupropion/Naltrexone on Caloric Intake by Rats Fed Palatable Diet

Of all the tested compounds, only KSK-73 administered at a dose of 10 mg/kg b.w. significantly reduced animals caloric intake over the 28-days of the experiment. The amount of calories consumed by the animals receiving KSK-59 or bupropion/naltrexone was comparable to the amount consumed by the control rats receiving i.p. vehicle and fed palatable feed. Results are shown in Figure 4a,b.

### 2.6. Influence of the Tested Compounds or Bupropion/Naltrexone on Plasma Total Cholesterol, HDL-Cholesterol, LDL-Cholesterol and Triglyceride Levels

No statistically significant differences in plasma total cholesterol, LDL cholesterol and HDL cholesterol were observed between all tested groups. Plasma triglyceride levels were statistically significantly higher in control rats fed palatable feed and in rats fed palatable feed and receiving KSK-59 compared to the rats fed standard feed. 

After multiple administrations of KSK-73 or bupropion/naltrexone to rats fed palatable feed their triglyceride plasma levels were comparable to the ones observed in rats from the control group fed standard feed. The results are shown in Figure 5a–d.

### 2.7. Glucose Tolerance after Treatment with the Tested Compounds or Bupropion/Naltrexone

Blood glucose levels at 30 and 60 min after the glucose load in control rats fed palatable feed were significantly higher compared to the glucose levels determined at the same time points in control rats fed standard feed. In the groups treated with KSK-59, KSK-73 at a dose of 10 mg/kg b.w. or bupropion/naltrexone glucose levels were significantly lower at 30 or/and 60 min after glucose load than in control animals fed palatable feed (Figure 6a–c). As shown in Figure 6d, the AUC was statistically significantly decreased by the treatment with KSK-59, KSK-73 or the reference compound compared to the control value observed in rats fed high-calorie feed.

### 2.8. Influence of Tested Compounds or Bupropion/Naltrexone on Plasma Leptin, Ghrelin and Adiponectin Levels

There were no significant changes in plasma adiponectin, leptin or ghrelin levels between rats from the control groups receiving different diets (standard or palatable) and i.p. vehicle (Figure 7a–c). Significantly lower level of adiponectin was observed in the plasma of rats fed palatable feed and treated with KSK-59 compared to the control rats fed palatable feed. The same was true for rats fed palatable feed and treated with bupropion/naltrexone. In this experiment the test compounds did not affect rats’ plasma leptin levels. Animals treated with bupropion/naltrexone had statistically significantly increased ghrelin levels compared to the rats that had access to the palatable products.

### 2.9. Effects on Visceral Illness via Measurement of Kaolin Intake

Animals treated with KSK-59 or KSK-73 at a dose of 10 mg/kg or bupropion/naltrexone did not consume more kaolin compared to the control group that received only vehicle and to the control group that received i.p. CuSO_4_. The decrease in body weight (vs the control group that received only vehicle) observed in the groups treated with the test compounds was similar to that seen in the control group that received i.p. CuSO_4_. Animals treated with the KSK-73 at a dose of 10 mg/kg b.w. also consumed fewer calories compared to the animals from the control group that received vehicle. Similarly reduced caloric intake was observed in the group that received i.p. CuSO_4_. Of note, the rats treated with bupropion/naltrexone consumed more calories than rats receiving CuSO_4_. Animals treated with the test compounds, CuSO_4_ or bupropion/naltrexone drank less water than ones from the control group that received only vehicle. Compared to the control group that received i.p. CuSO_4_, lower water intake was also noted in animals that received compounds KSK-59 or KSK-73. The results are shown in Figure 8a–d.

### 2.10. Influence of the Tested Compounds or Bupropion/Naltrexone on Spontaneous Activity

Compound KSK-59 at a dose of 10 mg/kg did not affect spontaneous activity after either first or chronic administrations. A statistically significant decrease in spontaneous activity was noted after a first administration of KSK-73, but no such changes were observed at the end of experiment (on the 27th day). Bupropion/naltrexone slightly decreased spontaneous activity after first i.p. administration. However, no change in spontaneous activity was observed after its chronic administrations. Results are shown in Figure 9a–f.

### 2.11. Influence of the Tested Compounds on Plasma Corticosterone, CRP and IL-6 Levels

There was no significant change in plasma corticosterone, CRP or IL-6 levels between rats from the control groups. 

In this experiment, the test compounds had no effect on rats’ plasma corticosterone or CRP levels. Plasma IL-6 levels in animals treated with bupropion/naltrexone were significantly higher compared to the ones observed in rats that received only vehicle and had access to the standard feed. Results are shown in Figure 10a–c.

### 2.12. Pharmacokinetic Analysis

Pharmacokinetic (PK) parameters calculated using non-compartmental analysis are summarized in Table 3. After i.p. administration both compounds reached maximal concentration (C_max_) at the first sampling point (5 min after administration) however the C_max_ of KSK-73 was much higher compared to KSK-59 (216 vs. 96 µg/L). Compound KSK-73 had longer half-life and lower clearance and as a consequence also the area under the curve extrapolated to infinity (total drug exposure across time) for KSK-73 was higher. Both compounds had quite high volume of distribution what might indicate their ability to freely cross biological barriers.

## 3. Discussion

In this study we performed preliminary pharmacological tests in order to determine the potential anti-obesity properties of two histamine H_3_ receptor antagonists. The influence of chronic i.p. administration on body weight, food intake, selected metabolic parameters and spontaneous activity of rats in the model of excessive eating of preferential feed was tested for compounds KSK-59 and KSK-73. 

In the presented model, experimental animals had continuous access to standard diet, which was additionally enriched with high-calorie products such as cheese, peanuts, chocolate or condensed milk. Such a model allows not only for the development of obesity and increase in the amount of fat in the peritoneum in the studied animals, but also for the development of certain metabolic disorders [11,12,18,19]. Already after the first week of the experiment, we noticed that animals from the control group that consumed palatable feed weighed significantly more than animals that consumed only standard feed. 

We observed significantly slower weight gain in the animals treated with the tested compounds at a dose of 10 mg/kg b.w. compared to the control animals fed palatable diet and administrated only vehicle (at a dose of 1 mg/kg b.w. compounds were inactive). Interestingly, compound KSK-73 administered at a dose of 1 mg/kg b.w. caused significantly greater weight gain compared to the changes determined in the control group fed palatable feed. This may be because at different doses, different mechanisms of action are dominant. The compound KSK-59 did not significantly affect weight gain when administered at a dose of 1 mg/kg b. w. which additionally suggests that KSK-73 and KSK-59 may act through yet additional, unknown to us mechanisms. Of note, treatment with KSK-73 at a dose of 10 mg/kg resulted even in a weight loss after the first administration. Additional studies that we conducted—monitoring of the animals’ spontaneous activity with a stress-free telemetry method—revealed that this effect could have been due to a reduction in activity after the first administration of KSK-73. The decrease in spontaneous activity could reduce the food intake by rats which consequently led to a slower body weight gain. Perhaps for this reason the sum of caloric intake in these animals was significantly lower than in the other groups. Such a change—decrease in spontaneous activity—was not observed after repeated administrations of KSK-73, however, body weight gain was still slower. That significantly shows that the influence on spontaneous activity in the initial phase of KSK-73 administration was not the only cause of slower weight gain.

Pharmacotherapy may lead to gastrointestinal side effects in the form of nausea and vomiting. When testing our compounds, we had to exclude such effects since the occurrence of gastrointestinal disturbances could cause a decrease of feed consumption by the experimental animals and lead to the false-positive results. Rodents do not have a vomiting reflex but may exhibit some feeding behavior, of ingesting non-nutritive substances such as kaolin clay, after a nausea-inducing stimulus. This behavior is termed as pica behavior [20,21]. In the study, in which rats had continuous access to kaolin clay, animals treated with KSK-59 or KSK-73 at a dose of 10 mg/kg b.w. consumed significantly less kaolin clay than rats from the control group that received i.p. CuSO_4_ (chemical substance causing nausea) [22]. These results indicate that the reduction in food intake during administration of KSK-73 was not associated with gastrointestinal disturbances.

In our research group, we are looking for compounds with anorectic activity among histamine H_3_ receptor antagonists. Testing histamine H_3_ receptor ligands in a rat model of excessive eating confirmed our assumptions that some compounds designed and synthesized in our research group may have potent anorectic effects and simultaneously improve selected metabolic parameters. In a previous paper, we published results for a compound 9 (KSK-94, structural analog of Abbott A-331440) that shows very strong affinity at histamine H_3_ receptors. It strongly prevented weight gain and reduced kcal intake in laboratory animals. It is noteworthy that KSK-94 also beneficially influenced some metabolic parameters by lowering plasma triglyceride and total cholesterol levels [12]. Of the two compounds tested in the present study, KSK-59 proved to be a more potent antagonist of the histamine H_3_ receptor. The compound KSK-73 showed a slightly lower activity, however still similar to thioperamide—the known antagonist of histamine H_3_ receptor. Our last work showed that the compounds with the strongest affinity for histamine H_3_ receptors were not the most active in in vivo models [11,12]. However, it is noteworthy that the compounds that most potently inhibited body weight gain in the rat model of excessive eating, were characterized by similar favorable pharmacokinetic parameters such as long half-life and large volume of distribution. Moreover, they had similar ability to cross the biological membranes determined in in vitro studies. This is also true for the compounds tested in this research. The more active in in vivo experiments KSK-73 had larger volume of distribution (216 vs. 152 L/kg), longer half-life (3.7 vs. 1.6 h) and reached higher concentrations in plasma. Although previously tested compounds inhibited weight gain, not all of them reduced caloric intake. It is, therefore, certain that it is not an anorectic effect (or not only) that plays a key role in the weight loss effect induced by administration of the histamine H_3_ receptor ligands.

We suspect that the anorectic activity as well as the metabolic activity of our compounds could be additionally related to a mechanism of action other than antagonism to histamine H_3_ receptors. Reports appear in the literature that the histamine H_3_ receptor ligands with the structure of piperazine derivatives may have an affinity at sigma-1 and sigma-2 receptors [23]. Determination of affinity and intrinsic activity for these two receptors should be included in a panel of histamine H_3_ ligand studies, and absence of such studies is a limitation of this experiment.

In all the studies we have carried out so far, we discovered reduction in the animals’ body weight gain after administration of the active compounds, along with a decrease in the amount of fat pads in the peritoneum [11,12]. This significantly and unequivocally shows activity of these compounds and benefits of such treatment. Administrations of KSK-73 also resulted in the reduction of the amount of intraperitoneal adipose tissue. Interestingly, this effect was even stronger than after i.p. injections of bupropion/naltrexone combination, which is currently registered for the treatment of overweight and obesity.

Very often obesity is accompanied by metabolic disorders such as elevated levels of glucose, triglycerides and cholesterol. In our experiment, we did not observe differences in plasma cholesterol levels between the experimental animals from all tested groups. This could have been due to the limited duration of our experiment, which did not last long enough to induce the above-mentioned metabolic disorders. Although, we observed that rats fed palatable feed had higher triglyceride levels than rats that had access only to standard feed. In addition, in our experiment, we noticed that elevated triglyceride levels were associated with increased amount of peritoneal fat. It is well known that triglycerides are part of adipose tissue and are a primary source of energy [24]. When energy intake exceeds the body’s needs triglycerides are stored in adipose tissue and simultaneously their blood concentration also increases. Thus, our results seem to be a logical consequence of this processes. Similar to the control group which were fed palatable feed, rats treated with KSK-59, had increased levels of plasma triglycerides that might be related to the increased levels of body fat in these groups. Levels of plasma triglyceride did not increase during treatment with the KSK-73, therefore, we selected this compound as especially worthy of attention and further research. 

In the excessive eating model used in this study, rats fed palatable feed had impaired glucose tolerance (control obese group) while treatment with either KSK-59 or KSK-73 caused significant improvement. It is generally known that impaired glucose tolerance may develop secondary to obesity [25], so these results, once again show the activity of histamine H_3_ receptor ligands as the compounds causing not only slower weight gain but also ameliorating some metabolic disorders in rats having the opportunity to overeat.

Previously we reported that other histamine H_3_ receptor antagonist—KSK-19—not only prevented weight gain in a mouse model of obesity, but also significantly improved glucose tolerance and insulin resistance [10]. Therefore, we plan to carry out similar tests (using a model of induced obesity and only after its initiation administer the test compounds) for our other compounds that work most favorably in the screening experiments on the model of excessive eating—including KSK-73.

Our study showed no significant differences in the plasma levels of leptin and adiponectin between the control groups fed standard and palatable feed. These adiponectins, secreted by adipose tissue function as a peripheral signal in the central regulation of energy balance [26]. Adiponectin influences a number of metabolic processes including glucose and fatty acid metabolism in liver and muscles [27,28]. Barnea et al. showed that reduced adiponectin RNA expression and thus reduced level of this hormone in adipose tissue were noted in a group of experimental animals that consumed fat-rich foods [29]. Interestingly, decreased adiponectin levels in adipose tissue did not lead to decreased plasma levels. This may suggest that in diet-induced obesity, low levels of adiponectin mRNA are not always associated with a concomitant reduction in plasma adiponectin levels [29,30,31]. In our experiment, we observed a decrease of adiponectin concentration in the plasma of rats treated with KSK-59 or bupropion/naltrexone. Thus, it seems necessary to investigate also the levels of adiponectin in adipose tissue, which is the main site of its production. 

Leptin is a hormone with a key role in food intake and body weight homeostasis. It takes part in the maintenance of body weight by inhibiting food intake and increasing energy expenditure. Leptin levels are directly proportional to the amount of body fat and therefore obesity is usually associated with hyperleptinemia and leptin resistance [32]. In our experiment we did not notice significant differences in the plasma levels of leptin however this could be due to the fact that experiment did not last long enough to induce leptin resistance. 

Ghrelin is a peptide hormone predominantly secreted by stomach. It has adipogenic and orexigenic properties, and therefore can increase food intake and body weight [33]. Plasma ghrelin concentrations have been reported to be reduced in obese subjects [34,35]. However, after weight loss, circulating levels of ghrelin, are increased, indicating a down regulation of ghrelin release as a result of energy excess in obesity [36]. In our study, only the combination of bupropion/naltrexone led to a significant increase in plasma ghrelin, which correlates with lower weight gain. However, taking into account that KSK-73 reduced caloric intake, it is rather clear that the total level of this hormone remained at a level comparable to that observed in the control group fed standard feed. 

Stress may be the most important factor influencing metabolism and feeding behavior leading to a short-term weight loss in both humans and rodents [37,38,39]. The acute stress response causes activation of the hypothalamic-pituitary-adrenal axis, which results in the release of glucocorticoids. Cortisol is released in response to stress in humans, whereas corticosterone is released in rodents [40]. In our experiment, corticosterone levels were assessed in experimental animals to rule out a stress factor that may contribute to weight loss. We observed no statistically significant differences in corticosterone levels in any of the experimental groups and excluded a stress factor as a potential cause of weight loss in the test animals.

It has been shown that mild inflammation and changes in the levels of circulating pro-inflammatory cytokines are observed not only in depression, schizophrenia, and diabetes, but also in eating disorders such as obesity or anorexia [41,42,43]. Some studies reported that weight loss is related to ongoing inflammation and results from the action of pro-inflammatory factors such as cytokines and prostaglandins on the nervous system [44]. Therefore, it seems that, on one hand, inflammation accompanies obesity, but on the other, it can also lead to weight loss. Therefore, as to exclude that fact, that weight loss in our experimental animals was caused by inflammation, we determined the levels of C-reactive protein and IL-6 in the plasma of animals at the end of chronic experiment. We observed no statistically significant differences in the levels of these inflammatory markers between any of the tested groups. The levels of inflammatory markers in the group fed standard feed was similar to the levels observed in the group fed high-calorie products. Apparently, in this model, the inflammation that coexists very often with obesity was not yet visible, but also the administration of tested compounds did not lead to its development. This again could be related to the short duration of the experiment. However, interestingly, high levels of IL-6 were observed in the group of rats receiving the bupropion/naltrexone combination, which possibly was related to adverse effects of this combination or its individual components. This issue remains to be clarified.

## 4. Materials and Methods

### 4.1. Drugs, Chemical Reagents and Other Materials

Tested compounds KSK-59: 1-(4-((6-(4-(pyridin-4-yl)piperazin-1-yl)hexyl)oxy) phenyl)ethan-1-one oxalate and KSK-73: 1-(4-((8-(4-(pyridin-4-yl)piperazin-1-yl)octyl) oxy)phenyl)ethan-1-one oxalate were synthesized at the Department of Technology and Biotechnology of Drugs, Faculty of Pharmacy, Jagiellonian University Medical College, Cracow, Poland. Identity and purity of the final product were assessed by NMR and LC-MS techniques (the minimum purity was more than 95%). For both pharmacokinetic and pharmacological studies KSK-K59 and KSK-73 (10 mg/kg b.w. or 1 mg/kg b.w. of rats) were suspended in 1% Tween 80 and the volume was adjusted to 10 mL/kg. Heparin was delivered by Polfa Warszawa S.A. (Warsaw, Poland), thiopental sodium was obtained from Sandoz GmbH, (Kundl, Austria), bupropion from TCI Co. (Fukaya, Japan) and naltrexone from Sigma-Aldrich (Munich, Germany).

The compounds were administrated i.p. to exclude the influence of special feeding on absorption and to avoid the exposure of animals to excessive stress associated with multiple administrations (multiple intragastric administrations are much more stressful for the animals than intraperitoneal).

### 4.2. In Vitro Studies

#### 4.2.1. Intrinsic Activity at Histamine H3 Receptor

Intrinsic activity studies were performed by two methods, Aequoscreen and Lance cAMP assays according to the manufacturer of the ready to use cells with stable expression of the H_3_ histamine receptor (Perkin Elmer, Waltham, MA, USA).

The Aequoscreen technology uses the recombinant cell lines with stable co-expression of apoaequorin and a GPCR as a system to detect activation of the receptor, following addition of an agonist, via the measurement of light emission. For measurement cells (frozen, ready to use) were thawed and resuspended in 10 mL of an assay buffer containing 5 μM of coelenterazine h. Then cells suspension was placed in a 10 mL Falcon tube, fixed onto a rotating wheel and incubated overnight at RT° in the dark. Cells were diluted with assay buffer to 5000 cells/20 µL. Agonistic ligands 2× (50 μL/well), diluted in assay buffer, were prepared in 1/2 area white polystyrene plates, and the 50 μL of cell suspension was dispensed on the ligands using the injector. The light emission was recorded for 20 s. Cells were incubated with antagonist for 15 min at RT°. Then 50 µL of agonist (3 × EC_80_ final concentration) was injected onto the mixture of cells and antagonist and the light emission was recorded for 20 s.

The LANCE Ultra cAMP assay as a homogeneous time-resolved fluorescence resonance energy transfer (TR-FRET) immunoassay is designed to measure cAMP produced upon modulation of adenylyl cyclase activity by GPCRs. The assay is based on the competition between the europium (Eu) chelate-labeled cAMP tracer and sample cAMP for binding sites on cAMP-specific monoclonal antibodies labeled with the ULight dye.

For measurement cells (frozen, ready to use) were thawed and resuspended in 4 mL of HBSS 1X. Then cells suspension was placed in a 15 mL Falcon tube and centrifuged for 10 min at 275× *g*. Pellet was resuspended in 1.5 mL HBSS 1X to determine cells concentration, and after next centrifugation cells were resuspended in stimulation buffer at appropriate concentration. An antagonist dose-response experiment was performed in 96-well 1/2 area plates using 3000 cells/well, 5 µM forskolin and 2 nM (R)-alpha-methylhistamine dihydrobromide as a reference agonist. Cell stimulation was performed for 30 min at RT°, and the agonist and antagonists were added simultaneously.

#### 4.2.2. PAMPA Assay

The reference compounds, well-permeable caffeine and low-permeable norfloxacin were obtained from Sigma-Aldrich (St. Louis, MO, USA). Used for this study, pre-coated PAMPA Plate System Gentest™ was provided by Corning (Tewksbury, MA, USA). The system consisted of 96-well receiver filter plate pre-coated with structured layers of phospholipids and a donor microplate. The assay was performed as described previously [17,45]. In brief, the tested compound KSK-59 and the reference compounds were diluted in the PBS buffer (pH 7.4) and applied into the plate at the final concentration of 200 μM. After 5 h of incubation at RT° the UPLC-MS spectrometry (Waters ACQUITY™ TQD system with the TQ Detector, Waters, Milford, MA, USA) method was used to estimate the quantity of compounds that penetrated from donor to acceptor wells. The permeability coefficients (*Pe*, cm/s) were calculated using the formulas provided by Corning.

### 4.3. Animals

The experiments were carried out on female Wistar rats with an initial body weight of 140–160 g. The animals were housed in plastic cages (2 rats per cage) at a constant room temperature of 22 ± 2 °C, with 12:12 h light/dark cycle. Water and food were available ad libitum. The randomly established experimental groups consisted of 6 rats for pharmacological studies and 3 rats for pharmacokinetic studies.

The experience related to this model suggests that greater differences in body weight changes and in selected metabolic parameters are observed in female rats [12,18,19,46]. Therefore, in preliminary studies, to make the effect more noticeable and in line with the principle of reducing the number of animals (with greater differences fewer animals per experimental group can be used to demonstrate statistical significance), female rats were selected. During the experiment the estrous cycle was not recorded.

All experiments were conducted in accordance with the Guide to the Care and Use of Experimental Animals and were approved by the Local Ethics Committee for Experiments on Animals of the Jagiellonian University in Krakow (Permission No: 185/2017, 220/2019 and 223A/2019).

### 4.4. Experimental Methods

#### 4.4.1. Effect of Tested Compound on Changes of Rats Body Weight and Calorie Intake in Non-Obese Rats Fed Palatable Diet (Model of Excessive Eating)

Female Wistar rats were housed in pairs. Two groups of 8 rats had access to diet consisting of milk chocolate with nuts, cheese, salted peanuts, and 7% condensed milk and simultaneously to a standard feed (Labofeed B, Morawski Manufacturer Feed, Łódź, Poland) for 4 weeks [11,18,46]. The water was available ad libitum. Palatable control group (palatable diet + vehicle) received vehicle (1% Tween 80, i.p.), while palatable test groups (palatable diet + compound) were injected i.p. tested compounds at the dose of 10 or 1 mg/kg b.w./day suspended in 1% Tween 80 or the reference compound bupropion/naltrexon. The combination of bupropion at a dose of 20 mg/kg b.w./day and naltrexone at a dose of 1 mg/kg b.w./day was also suspended in 1% Tween 80 and administered i.p. Weight was evaluated daily. Calorie intake was evaluated ones per week. On the 31st day, 20 min after i.p. administration of heparin (5000 units/rat) and thiopental (70 mg/kg b.w.), animals were sacrificed and peritoneal fat pads were collected (Figure 11).

Palatable diet contained: 100 g peanuts—612 kcal; 100 mL condensed milk—131 kcal; 100 g milk chocolate—529 kcal; 100 g cheese—325 kcal. Standard diet contained 100 g feed—280 kcal.

#### 4.4.2. The Effect of Tested Compounds on Spontaneous Activity in Non-Obese Rats Fed Palatable Diet

The spontaneous activity of rats was measured on the 1st and 27th day of the treatment with a special RFID-system—TraffiCage (TSE-Systems, Germany). The animals were subcutaneously implanted with transmitter identification (RFID), which enabled the presence and time spent in different areas of the cage to be counted and then the data were grouped in a special computer program [47,48].

#### 4.4.3. Influence on Visceral Illness via Measurement of Kaolin Intake (Pica Behavior)

The method was based on the works by Takeda et al. and Yamamoto et al. [21,22] with minor modification [18]. The experiment lasted for five days. In addition to free access to feed, animals had also free access to a white kaolin. For the first few days, the animals were accustomed to the presence of kaolin in their cages. On the 4th day, either KSK-59, KSK-73 (10 mg/kg b.w.), a vehicle (negative control group), or a solution of CuSO_4_ (6 mg/kg b.w.—1/3 LD_50_; LD_50_ = 18 mg/kg for a rat at this route of administration; positive control group) were administered i.p. The amounts of consumed food, water and kaolin were determined after 24 h. Moreover, animals were weighed before the compounds administration and 24 h after.

#### 4.4.4. Glucose Tolerance Test 

The test was performed on the 29th day of the experiment. After twenty-eight administrations of the test compounds, food was discontinued for 20 h and then glucose tolerance was tested. Glucose (1 g/kg b.w.) was administrated i.p. [18,19]. Blood samples were taken at the time points: 0 (before glucose administration), 30, 60 and 120 min after administration from the tail vein [11]. Glucose levels were measured with glucometer (ContourTS, Bayer, Leverkusen, Germany, test stripes: ContourTS, Ascensia Diabetes care Poland Sp. z o.o., Poland, REF:84239666). The area under the curve (AUC) was calculated using the trapezoidal rule.

### 4.5. Influence of KSK-59, KSK-73 or Bupropion/Naltrexone on Corticosterone, Total Cholesterol, HDL-Cholesterol, LDL-Cholesterol, Triglyceride, IL-6, Ghrelin, Adiponectin, Leptin and C-Reactive Protein Levels in Plasma

On the 31st day of the experiment, 20 min after i.p. administration of heparin (5000 units/rat) and thiopental (70 mg/kg b.w.), blood was collected from the left carotid artery and then centrifuged at 600× *g* (15 min, 4 °C) in order to obtain the plasma. To determine total cholesterol, HDL-cholesterol, LDL-cholesterol or triglyceride levels in plasma, standard enzymatic and spectrophotometric tests (Biomaxima S.A. Lublin, Poland) were used. The ELISA Kit (Cayman Chemical, USA) assay was used to determine corticosterone levels in plasma. To determine IL-6, ghrelin, adiponectin, leptin and C-reactive protein levels in plasma, standard ELISA Kit (Bioassay Technology Laboratory, Shanghai, China) was used. 

### 4.6. Pharmacokinetic Studies

Six rats divided into two experimental groups were used in pharmacokinetic experiments. Three days before the experiment, rats’ jugular vein was cannulated allowing for the multiple blood sampling from the single animal. Investigated compounds suspended in 1% Tween 80 were administered i.p. at the single dose of 10 mg/kg. Blood samples (approximately 300 µL) were collected to Eppendorf tubes containing heparin at 5, 15, 30, 60, 120, 240 and 360 min after dosing. Plasma was harvested by centrifuging at 5000× *g* for 10 min and stored at −30 °C until bioanalysis. 

Plasma concentrations of KSK-59 or KSK-73 were measured by liquid chromatography tandem mass spectrometry (LC-MS/MS) method. Samples (50 μL) were deproteinized at the ratio of 1:3 (*v*/*v*) with acetonitrile containing an internal standard (IS—pentoxifiline), briefly vortexed and then centrifuged for 10 min at the speed of 8000× *g* (Eppendorf miniSpin centrifuge). The supernatant was transferred into the autosampler vials and a sample volume of 10 μL was injected into LC-MS/MS system. 

Chromatographic separation was carried out on XBridge™ C18 (3 × 50 mm, 5 µm, Waters, Dublin, Ireland) analytical column using the HPLC system Agilent 1100 (Agilent Technologies, Waldbronn, Germany). The mobile phase containing 0.1% formic acid in acetonitrile and in water was run at 0.3 mL/min in the gradient mode. Mass spectrometric detection was performed on an Applied Biosystems MDS Sciex (Concord, ON, Canada) API 2000 triple quadrupole mass spectrometer. Electrospray ionization (ESI) in the positive ion mode was used for ion production. The tandem mass spectrometer was operated at unit resolution in the selected reaction monitoring mode (SRM), monitoring the transitions of the protonated molecular ions *m*/*z* 382 to 107 for KSK-59, *m*/*z* 410 to 107 for KSK-73 and *m*/*z* 279 to 181 for IS. Data acquisition and processing were accomplished using the Applied Biosystems Analyst version 1.6 software. The calibration curves were constructed by plotting the ratio of the peak area of the studied compound to IS versus drug concentration and generated by weighted (1/x·x) linear regression analysis. The validated quantitation ranges were from 1 to 2000 ng/mL. Calculated accuracy and precision were within the ranges proposed by guidelines for bioanalytical methods validation (FDA, EMA). No significant matrix effect was observed and there were no stability related problems during the routine analysis of the samples.

Pharmacokinetic parameters were calculated by employing a non-compartmental approach, using Monolix version 2019R1 (Antony, France: Lixoft SAS, 2019) software. The area under the mean plasma concentration versus time curve extrapolated to infinity (AUC_0-inf_) was estimated using the log/linear trapezoidal rule. AUMC_0-inf_ was estimated by calculation of total area under the first-moment curve by combining trapezoid calculation of AUMC_0-t_ and extrapolated area. The mean residence time (MRT) was calculated from AUMC_0-inf_/AUC_0-inf_. The terminal rate constant (λ_z_) was calculated by log-linear regression of the drug concentration data in the terminal phase and the terminal half-life (t_1/2_) were calculated as 0.693/λ_z_. The clearance (CL/F) was estimated from the administered dose divided by AUC_0-inf_. The apparent volume of distribution during terminal phase (V_z_/F) was calculated from (CL/F)/λ_z_.

### 4.7. Statistical Analysis

Statistical calculations were performed using GraphPad Prism 6 program. Results are presented as arithmetic means with a standard error of the mean. The normality of data sets was determined using the Shapiro–Wilk test. Statistical significance was calculated using one-way ANOVA, Tukey post hoc or two-way ANOVA, Tukey post hoc (body weight) or Bonferroni post hoc (spontaneous activity). Differences were considered statistically significant at: * *p* ≤ 0.05, ** *p* ≤ 0.01, *** *p* ≤ 0.001.

## 5. Conclusions

In our work, we demonstrated that the histamine H_3_ receptor antagonists KSK-59 and KSK-73 administered i.p. at a dose of 10 mg/kg b.w. prevent weight gain in a rat model of excessive eating. Interestingly, animals treated with KSK-73 and fed palatable feed gained weight to the similar extent as the animals from the control group that received vehicle and had free access to the standard feed only. Compound KSK-73 reduced the caloric intake of the experimental animals indicating that it has an anorectic effect. We also observed significantly lower amount of intraperitoneal adipose tissue in animals treated with KSK-73 compared to a control group of rats fed palatable feed and receiving vehicle. Additionally, in animals from the chronic experiment, we investigated the effects of the test compounds on the plasma levels of selected metabolic parameters; however, for the most part, we did not observe significant changes in the individual assays. The reason for this could be that our experiment did not last long enough to induce such changes. However, it should be emphasized that the compounds we studied are potent H_3_ histamine receptor antagonists with proven efficacy in preventing weight gain in a rat model of excessive eating. 

## Figures and Tables

**Figure 1 pharmaceuticals-14-01080-f001:**
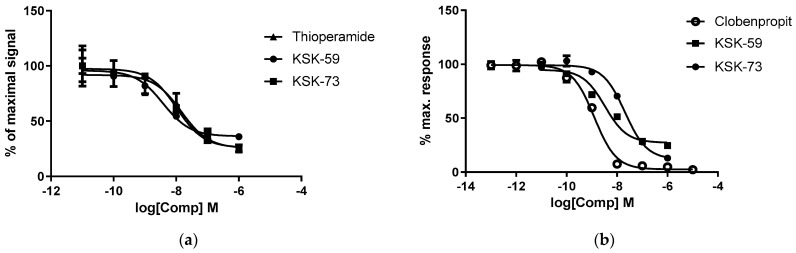
The intrinsic activity at H_3_ histamine receptor of the tested compounds presented as a concentration-dependence curves. (**a**) Lance cAMP, (**b**) Aequoscreen. Obtained values (%) are expressed as percent of the action of full agonist (*R*)-alpha-methylhistamine at the dose of EC_80_ (100%).

**Figure 2 pharmaceuticals-14-01080-f002:**
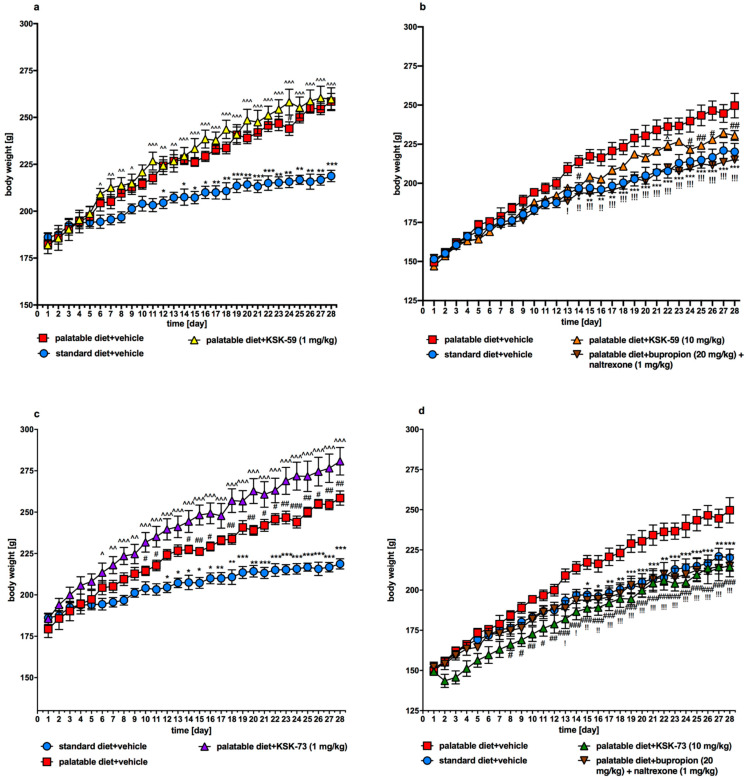
Body weight throughout the administration of the tested compounds or bupropion/naltrexone. (**a**) KSK-59 (1 mg/kg), (**b**) KSK-59 (10 mg/kg) and bupropion/naltrexone (10/1 mg/kg), (**c**) KSK-73 (1 mg/kg), (**d**) KSK-73 (10 mg/kg) and bupropion/naltrexone (10/1 mg/kg). Results are expressed as means ± SEM, *n*  =  6. Multiple comparisons were performed by two-way ANOVA, Tukey’s post-hoc tests. * Significant against control group fed standard diet vs. control group fed palatable diet; ^ significant against tested compound administered group vs. control group fed standard diet; # significant against tested compound or bupropion/naltrexone administered group vs. control group fed palatable diet; ! significant against bupropion/naltrexone administered group vs. control group fed palatable diet; *, ^, #, ! *p* < 0.05, **, ^^, ##, !! *p* < 0.01, ***, ^^^, ###, !!! *p* < 0.001.

**Figure 3 pharmaceuticals-14-01080-f003:**
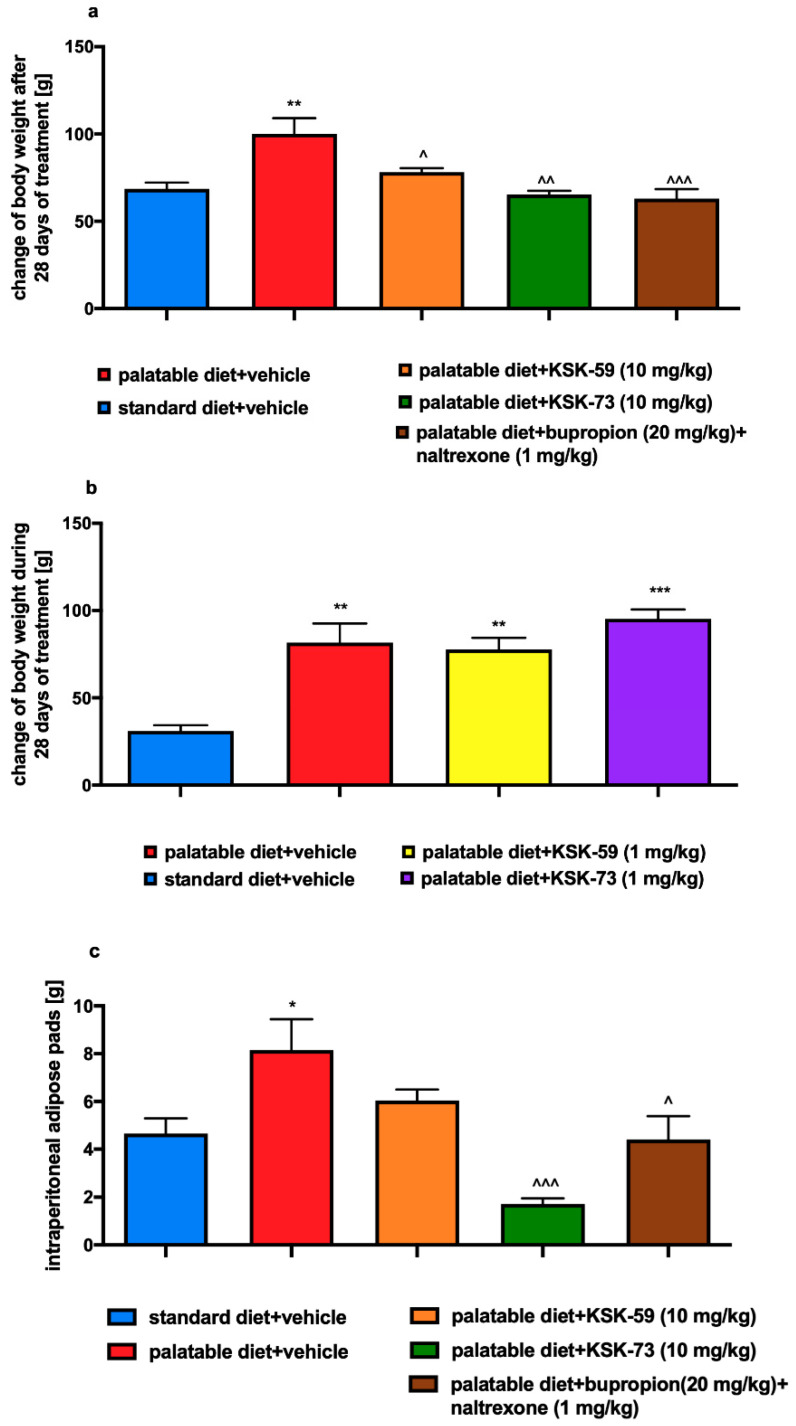
Cumulative changes in body weight and mass of adipose pads. (**a**,**b**) Cumulative changes in body weight. (**c**) Mass of adipose pads at the end of experiment. Results are expressed as means ± SEM, *n* = 6. Comparisons were performed by one-way ANOVA, Tukey’s post hoc tests. * Significant against control rats fed standard diet; ^ significant against control rats fed palatable diet; ^, * *p* < 0.05, **, ^^ *p* < 0.01, ***, ^^^ *p* < 0.001.

**Figure 4 pharmaceuticals-14-01080-f004:**
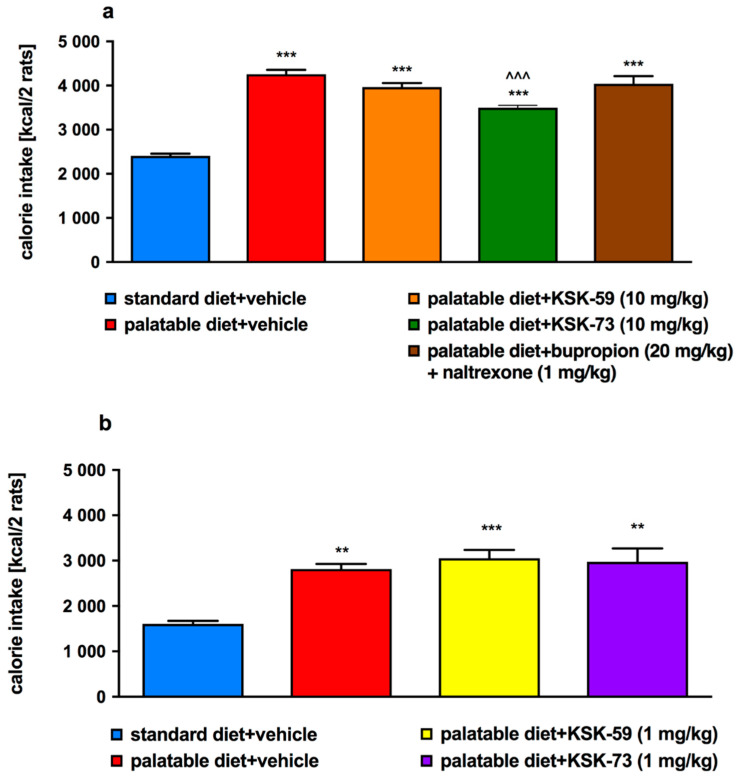
Effect of administration of the tested compounds or bupropion/naltrexone on calorie intake. The amount of calories consumed by rats treated with test compounds KSK-59 or KSK-73 at a dose of 1 mg/kg b.w. (**a**) or a dose of 10 mg/kg b.w. (**b**) compared to control groups. Results are expressed as means ± SEM, *n* = 6. Comparisons were performed by one-way ANOVA, Tukey’s post hoc tests. * Significant against control rats fed standard diet; ^ significant against control rats fed palatable diet; ** *p* < 0.01, ***, ^^^ *p* < 0.001.

**Figure 5 pharmaceuticals-14-01080-f005:**
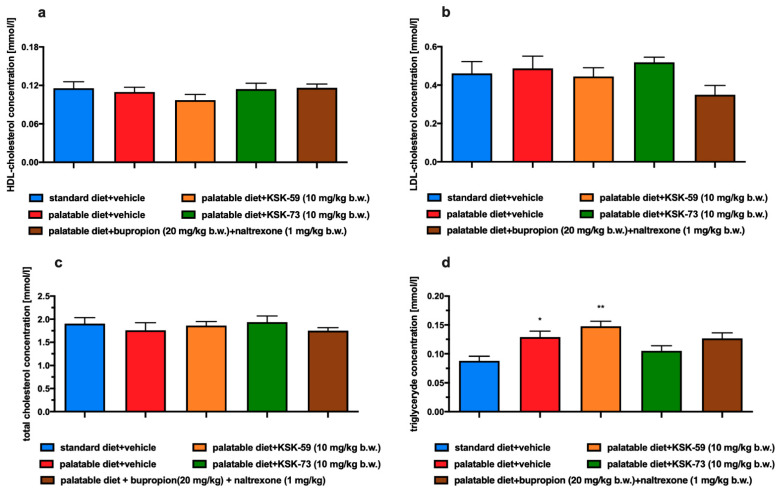
Effect of the tested compounds or bupropion/naltrexone administration on plasma levels of: (**a**) total cholesterol, (**b**) LDL cholesterol, (**c**) HDL-cholesterol, (**d**) triglyceride. Results are expressed as means ± SEM, *n* = 6. Comparisons were performed by one-way ANOVA, Tukey’s post hoc test. * Significant against control rats fed standard diet; * *p* < 0.05, ** *p* < 0.01.

**Figure 6 pharmaceuticals-14-01080-f006:**
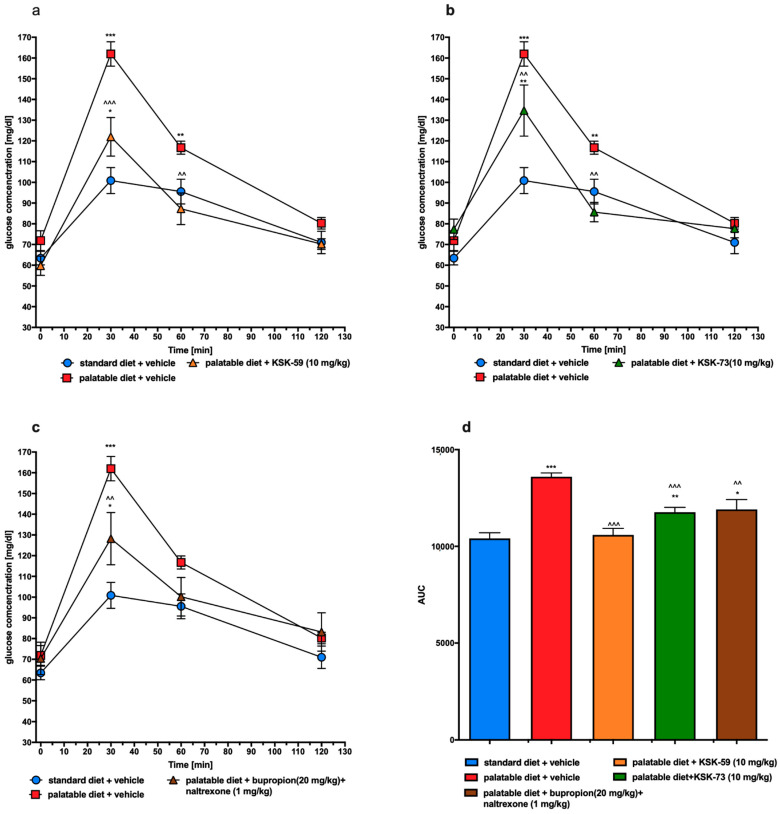
Glucose tolerance test. Results are expressed as means ± SEM, *n* = 6. (**a**–**c**) Intraperitoneal glucose tolerance test (IPGTT)—multiple comparisons were performed by two-way ANOVA, Tukey’s post-hoc tests. (**d**) Area under the curve of IPGTT—comparisons were performed by one-way ANOVA, Tukey’s post hoc test. * Significant against control rats fed standard diet; ^ significant against control rats fed palatable diet; * *p* < 0.05, **, ^^ *p* < 0.01, ***, ^^^ *p* < 0.001.

**Figure 7 pharmaceuticals-14-01080-f007:**
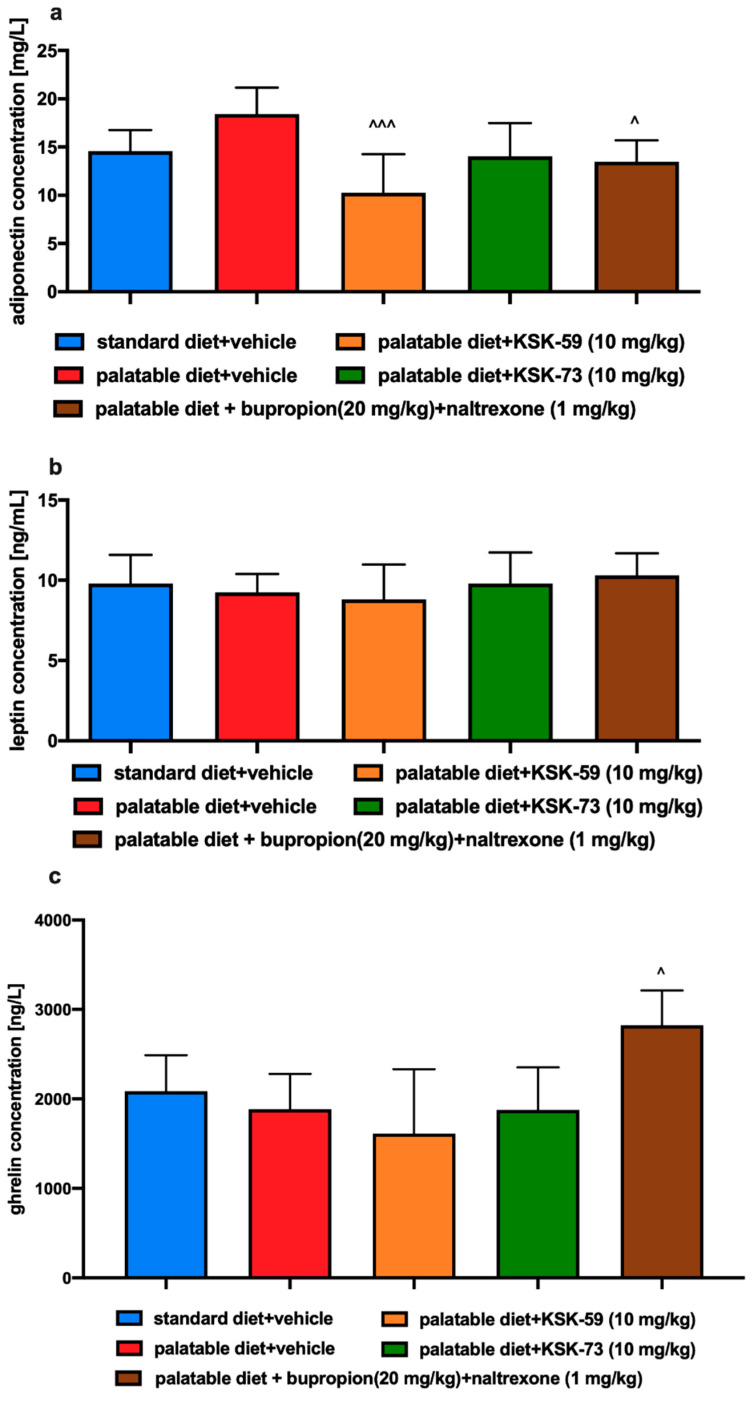
Effect of administration of the tested compound or bupropion/naltrexone on levels in plasma of: (**a**) adiponectin, (**b**) leptin, and (**c**) ghrelin. Results are expressed as means ± SEM, *n* = 6. Comparisons were performed by one-way ANOVA, Tukey’s post hoc test; ^ significant against control rats fed palatable diet; ^ *p* < 0.05, ^^^ *p* < 0.001.

**Figure 8 pharmaceuticals-14-01080-f008:**
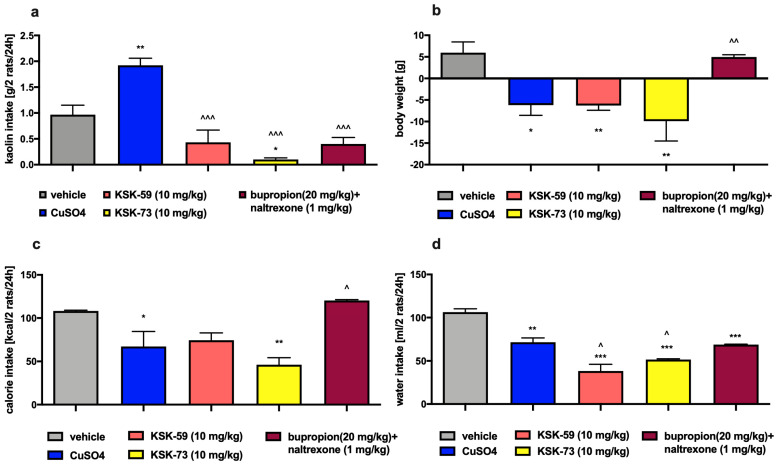
Effect of single administration of the tested compounds or bupropion/naltrexone on kaolin intake (**a**), body weight (**b**), calorie intake (**c**) and water intake (**d**) in female Wistar rats in the model pica behavior. Results are expressed as means ± SEM, *n* = 6. Comparisons were performed by one-way ANOVA, Tukey’s post hoc test. * Significant against control rats; ^ significant against control rats treated with CuSO_4_; *, ^ *p* < 0.05, **, ^^ *p* < 0.01, ***, ^^^ *p* < 0.001.

**Figure 9 pharmaceuticals-14-01080-f009:**
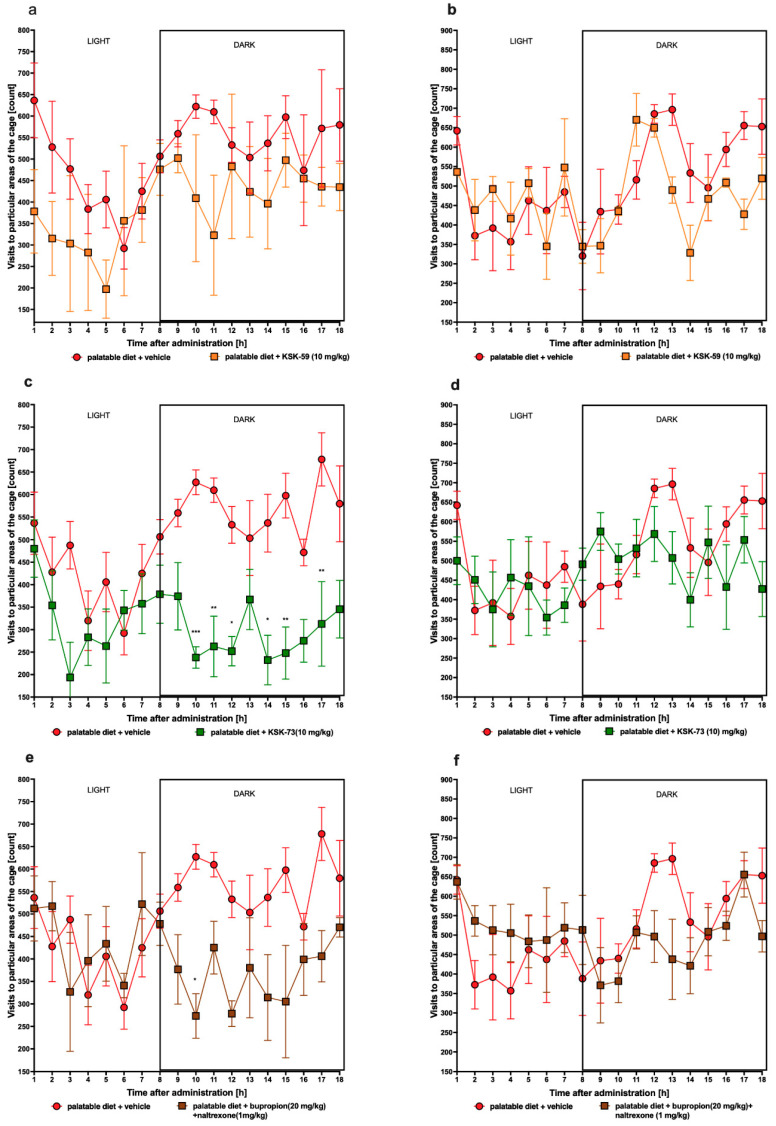
Spontaneous activity after the 1st and 27th administration of the tested compounds. Results are expressed as means ± SEM, *n* = 6. Comparisons were performed by two-way ANOVA, Bonferroni’s post hoc tests; (**a**) control group fed palatable diet and after 1st administration of KSK-59; (**b**) control group fed palatable diet and after 27th administration of KSK-59; (**c**) control group fed palatable diet and after 1st administration of KSK-73; (**d**) control group fed palatable diet and after 27th administration of KSK-73; (**e**) control group fed palatable diet and after 1st administration of bupropion/naltrexone; (**f**) control group fed palatable diet and after 27th administration of bupropion/naltrexone; * *p* < 0.05, ** *p* < 0.01, *** *p* < 0.001.

**Figure 10 pharmaceuticals-14-01080-f010:**
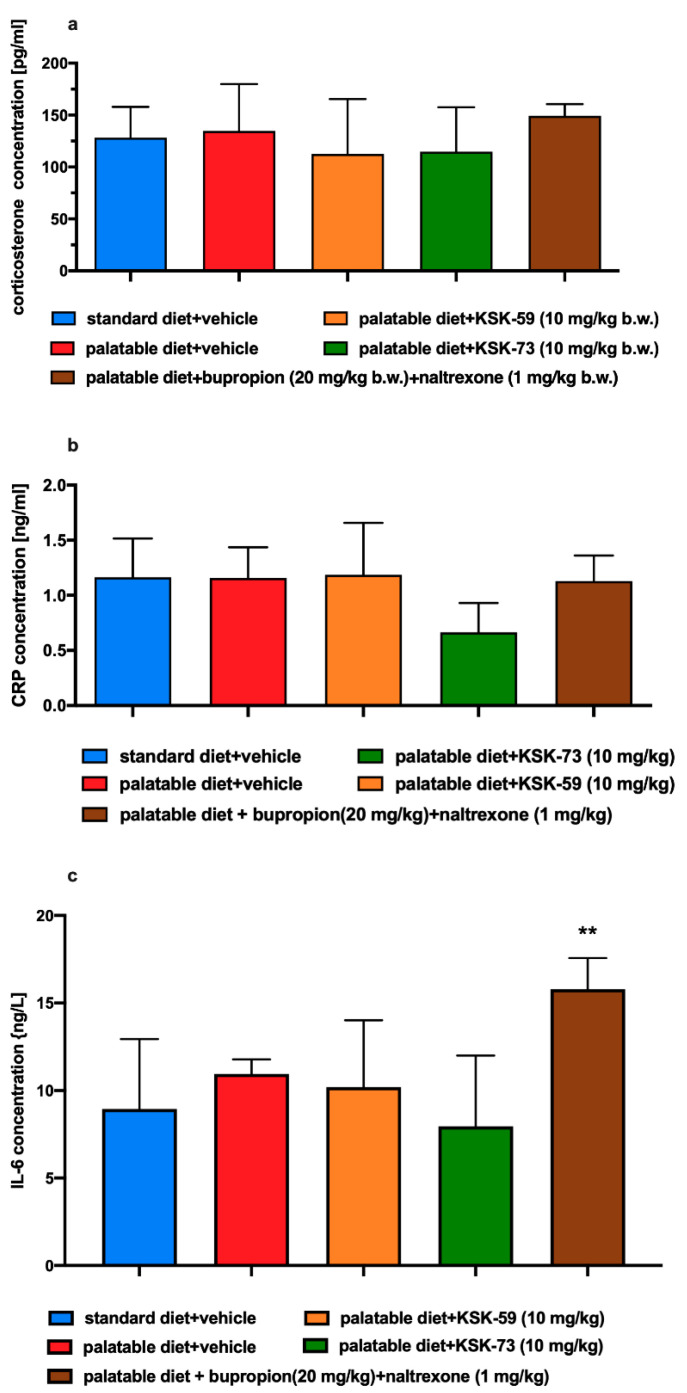
Effect of an administration of the tested compounds or bupropion/naltrexone on plasma levels of: (**a**) corticosterone, (**b**) CRP, and (**c**) IL-6. Results are expressed as means ± SEM, *n* = 6. Comparisons were performed by one-way ANOVA, Tukey’s post hoc test; * significant against control rats fed standard diet; ** *p* < 0.01.

**Figure 11 pharmaceuticals-14-01080-f011:**
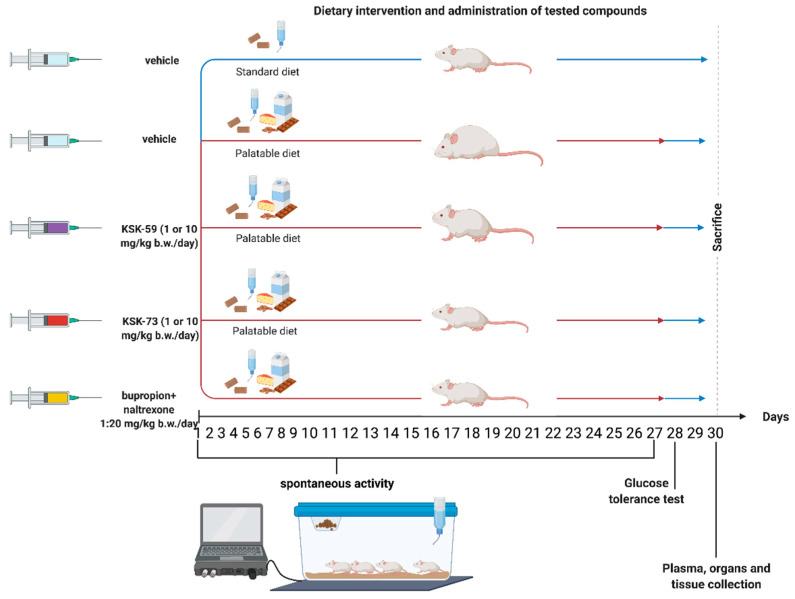
Scheme of the experiment.

**Table 1 pharmaceuticals-14-01080-t001:** The IC_50_ values for antagonist dose-response with reference agonist ((R)-alpha-methylhistamine) at a final concentration equivalent to the EC_80_. Obtained by two methods Lance cAMP and Aequoscreen.

Compound	Lance cAMP IC_50_ [nM]	Aequoscreen IC_50_ [nM]
(*R*)-alpha-methylhistamine	1.05 ± 0.1	10.6 ± 2.2
Clobenpropit	-------------	1.2 ± 0.3
Thioperamide	14.52 ± 3.2	---------------
KSK-59	3.79 ± 0.4	3.31 ± 3.2
KSK-73	10.47 ± 1.5	20.62 ± 4.5

**Table 2 pharmaceuticals-14-01080-t002:** The results obtained in PAMPA.

Compound	*Pe*^a^ (10^−6^ cm/s) ± SD
CFN ^b^	12.22 ± 0.94
NFX ^c^	0.056 ± 0.01
KSK-59	4.35 ± 0.93
KSK-73 ^d^	6.22 ± 2.3

*n* = 3, ^a^ permeability coefficient, ^b^ caffeine (CFN)—well-permeable drug, ^c^ norfloxacin (NFX)—low permeable drug, ^d^ published previously [17].

**Table 3 pharmaceuticals-14-01080-t003:** Estimated pharmacokinetic parameters (non-compartmental analysis) of the investigated compounds calculated from the mean rat plasma concentration values (*n* = 3) after single i.p. administration at a dose of 10 mg/kg.

Parameter	KSK-59	KSK-73
C_max_ [µg/L]	96	216
t_max_ [h]	0.083	0.083
λ_z_ [h^−1^]	0.44	0.19
t_0.5λz_ [h]	1.58	3.7
CL_S_/F [L/h/kg]	66.92	40.51
AUC_0-inf_ [mg∙h/L]	537.96	888.49
V_z_/F [L/kg]	152.22	216
MRT [h]	2.04	4.2

C_max_—maximal concentration, t_max_—time to reach maximal concentration, λ_z_—terminal elimination rate constant, t_0.5λz_—half-life, CL_s_—clearance, AUC_0-inf_—area under the concentration-time curve extrapolated to infinity, V_z_—volume of distribution, MRT—mean residence time, F-bioavailability.

## Data Availability

Data is contained within the article.

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
