# Peer review of "Histamine H3 Receptor Ligands—KSK-59 and KSK-73—Reduce Body Weight Gain in a Rat Model of Excessive Eating"

_pharmaceuticals, 2021, doi:10.3390/ph14111080_

Round 1

Reviewer 1 Report

In this paper, the results of preclinical studies on the effects of two novel antagonists of histamine H3 receptors are presented. The influence of a chronic (28days) administration of the compounds KSK-59 and KSK-73 on body weight, food intake, some metabolic parameters, and spontaneous activity in the rat model of obesity evoked by an excessive high-calorie feed intake was evaluated. Both compounds prevented weight gain, reduced adipose tissue deposits, and improved glucose tolerance, the compound KSK-73 being more effective what makes it a promising candidate for obesity treatment. The study design and research methods employed are sound. Obtained data are thoroughly discussed in the context of literature.

Minor Suggestion:

line 38: it is advisable to add  stronger      (...its stronger anorectic effect)

Author Response

In this paper, the results of preclinical studies on the effects of two novel antagonists of histamine H3 receptors are presented. The influence of a chronic (28days) administration of the compounds KSK-59 and KSK-73 on body weight, food intake, some metabolic parameters, and spontaneous activity in the rat model of obesity evoked by an excessive high-calorie feed intake was evaluated. Both compounds prevented weight gain, reduced adipose tissue deposits, and improved glucose tolerance, the compound KSK-73 being more effective what makes it a promising candidate for obesity treatment. The study design and research methods employed are sound. Obtained data are thoroughly discussed in the context of literature.

Minor Suggestion:

line 38: it is advisable to add  stronger      (...its stronger anorectic effect)

Authors’ answer:

Our studies showed an anorectic effect (reduction in the amount of kcal intake) only for the compound KSK-73 administered at a dose of 10 mg/kg b.w. (Figure 4a), therefore we think that addition of the word  "stronger" might be misleading.

Reviewer 2 Report

Mika et al wrote a very interesting paper about new anti-obesity agents acting on histamine H3 receptors. The paper is well written and easy-readable, although there are some aspects that need to be improved.

I suggest to add in the introduction (line 56-59) if the drugs are been approved by EMA or FDA.

In Results section: 
2.1: it will be better to add a phrase justifying why the two tests give such different values for IC50 for the same compound.
2.2: specify if the parameters obtained with PAMPA test are compatible with BEE crossing for the molecules (in the introduction the anti-obesogenic activity of H3 antagonists is linked to action in SNC, so it’s necessary to prove molecule can cross the blood/brain barrier; eventually doing another experiment if possible)
2.12: add the error as SD or SEM to the values in table 3.
Discussion is well written but misses a part of proposed mechanism of action of the compound, that can explain why H3 antagonism can cause body weight reduction; especially considering that no variation was observed in hormones concentrations in plasma. Moreover, it will be necessary to write a paragraph commenting the results obtained for body weight when the two compounds are administered at the dosage of 1 mg/kg (observed increase in body weight when compared to veh-treated animals).

About methods section:
It’s necessary to justify the choice of using female rats, and specify if during the experiment there was record of estrous cycle. Moreover, statistical analysis that has to be performed is two-way ANOVA for measurements contained in fig 3-8; since there are two variables (diet and treatment). 

Other suggestions may be about figures: 
Fig 1: it would be best to include buproprion/naltrexone curve in fig1a-d; to better evidence the difference between approved treatment and experimental one. In this way, it will be possible to enlarge the graphics, that in this figure are not easy-readable. In each graph, including the ones in other figures, I suggest when possible to magnify significances. Also, it would be interesting to add statistical analysis comparing proposed treatment with H3 antagonist and buproprion/naltrexone treatment. 
Fig 2: explain why figure 2b presents different values for change of body weight when compared with fig 2a in vehicle group exposed to standard and palatable diet, or specify they come from different experiments. Same in fig. 3 and 4.
Fig 3: fig3a misses a significance described in results line 194-195. 
